# Targeted Therapies in Advanced Biliary Tract Cancer: An Evolving Paradigm

**DOI:** 10.3390/cancers12082039

**Published:** 2020-07-24

**Authors:** Sakti Chakrabarti, Mandana Kamgar, Amit Mahipal

**Affiliations:** 1Department of Hematology-Oncology, Medical College of Wisconsin, 8701 Watertown Plank Road, Milwaukee, WI 53226, USA; schakrabarti@mcw.edu (S.C.); mkamgar@mcw.edu (M.K.); 2Department of Medical Oncology, Mayo Clinic, 200 First Street SW, Rochester, MN 55905, USA

**Keywords:** biliary tract cancer, cholangiocarcinoma, pemigatinib, ivosidenib, gallbladder cancer, targeted therapy, molecular profiling, next-generation sequencing, circulating tumor DNA (ctDNA)

## Abstract

Biliary tract cancers (BTCs) are a heterogeneous group of adenocarcinomas that originate from the epithelial lining of the biliary tree. BTCs are characterized by presentation with advanced disease precluding curative surgery, rising global incidence, and a poor prognosis. Chemotherapy is the mainstay of the current treatment, which results in a median overall survival of less than one year, underscoring the need for novel therapeutic agents and strategies. Next-generation sequencing-based molecular profiling has shed light on the underpinnings of the complex pathophysiology of BTC and has uncovered numerous actionable targets, leading to the discovery of new therapies tailored to the molecular targets. Therapies targeting fibroblast growth factor receptor (FGFR) fusion, isocitrate dehydrogenase (IDH) mutations, the human epidermal growth factor receptor (HER) family, DNA damage repair (DDR) pathways, and *BRAF* mutations have produced early encouraging results in selected patients. Current clinical trials evaluating targeted therapies, as monotherapies and in combination with other agents, are paving the way for novel treatment options. Genomic profiling of cell-free circulating tumor DNA that can assist in the identification of an actionable target is another exciting area of development. In this review, we provide a contemporaneous appraisal of the evolving targeted therapies and the ongoing clinical trials that will likely transform the therapeutic paradigm of BTC.

## 1. Introduction

Biliary tract cancers (BTCs) are a heterogeneous group of adenocarcinomas that originate in the epithelial lining of the bile ducts and gallbladder. Based on the anatomical location, BTCs are classified into two major categories—cholangiocarcinoma (CCA), which arises from the bile ducts, and gall bladder carcinoma (GBC). CCAs are subdivided into intrahepatic CCA (iCCA) and extrahepatic CCA (eCCA), with the eCCA further split into perihilar (Klatskin’s tumor) and distal CCA [1]. Each of these anatomic subgroups have their unique natural history and treatment nuances [1].

BTCs, although relatively uncommon in the United States, with 23,000 new cases diagnosed annually [2], constitute approximately 3% of all gastrointestinal malignancies [3]. Advanced BTC has a dismal prognosis, with a five-year survival rate of less than 10% [4]. The majority of patients with BTC (>65%) present with unresectable disease, and patients who undergo potentially curative surgery experience a high rate of relapse [5,6], underscoring the importance of systemic therapy. The current standard therapy of advanced BTC primarily consists of systemic chemotherapy, which results in a median overall survival (OS) of approximately 12 months [7]. However, targeted therapies are emerging as a promising treatment option, encouraged by the recent Food and Drug Administration (FDA) approval of pemigatinib for treatment of adults with chemotherapy-refractory CCA, i.e., advanced CCA harboring a fibroblast growth factor receptor (*FGFR*) 2 fusion.

As the understanding of the genomic landscape of BTC is growing, therapies targeting actionable alterations are being evaluated in numerous clinical trials, often combined with chemotherapy, both in frontline and chemotherapy-refractory settings [8]. Herein, we review the recent advancements and the emerging novel targeted therapies for BTC that will likely improve patient outcomes in the near future.

## 2. Advanced BTC: Current Standard of Care

Advanced BTC is defined as metastatic or unresectable tumors that are not amenable to local therapy with curative intent. The current standard treatment for patients with advanced BTC primarily consists of palliative chemotherapy. In the 1990s, gemcitabine was the only available palliative chemotherapy [9], which was later replaced by cisplatin-gemcitabine combination therapy after the ABC-02 trial results were published [7]. The ABC-02 trial was a phase III trial in which 410 patients with advanced BTC were randomized to receive either gemcitabine or a combination of cisplatin plus gemcitabine. The study reported a significant overall survival (OS) improvement with the cisplatin-gemcitabine doublet compared to single-agent gemcitabine (11.7 vs. 8.1 months; hazard ratio (HR), 0.64; 95% confidence interval (CI), 0.52 to 0.80; *p* < 0.001). However, gemcitabine monotherapy continues to be a viable option for patients with poor performance status or renal dysfunction who may not derive a benefit from the doublet [10]. Several studies have reported encouraging preliminary results in the first-line setting by adding a third chemotherapeutic drug to the cisplatin-gemcitabine backbone [11,12]. Shroff et al. [12] conducted a phase II trial with a triplet regimen consisting of cisplatin, gemcitabine, and nab-paclitaxel in therapy-naive patients with advanced BTC and reported an overall response rate (ORR) of 45%, a median progression-free survival (PFS) of 11.8 months (vs. eight months in ABC-02), and a median OS of 19.2 months (vs. 11.7 months in ABC-02). The ongoing phase III SWOG 1815 trial (ClinicalTrials.gov Identifier: NCT03768414) is evaluating this triplet regimen against the gemcitabine and cisplatin combination in newly diagnosed patients with advanced BTC.

Approximately 15–25% of patients are fit enough to receive second-line therapy [13,14,15]. Several retrospective studies [13,14,15] and a randomized phase III study (ABC-06) [16] have suggested a benefit of second-line chemotherapy following progression on cisplatin and gemcitabine. The standard second-line therapies include FOLFOX (5-FU, leucovorin, and oxaliplatin) [16], capecitabine plus irinotecan (XELIRI) [17], irinotecan monotherapy [17], and fluoropyrimidine (with 5-FU) monotherapy [13]. The recently reported phase III ABC-06 [16] study demonstrated a modest survival benefit with modified FOLFOX over active symptom control following progression on cisplatin and gemcitabine, with a median OS of 6.2 vs. 5.3 months, respectively (adjusted HR, 0.69; *p* = 0.03), establishing FOLFOX as a preferred treatment regimen in the second-line setting. However, a recent retrospective analysis of two large multicenter prospective cohorts suggested that monotherapy with 5-FU is as active as 5-FU doublets [18]. Chemotherapy intensification with the addition of irinotecan to FOLFOX (FOLFIRINOX), studied in a phase II trial in previously treated patients, reported a modest ORR, median PFS, and OS of 10%, 6.2, and 10.7 months, respectively [19]. A phase II study in refractory patients with a newer chemotherapeutic agent, TAS-102 (trifluridine and tipiracil), reported a median PFS and OS of 3.8 and 6.1 months, respectively, but no objective responses [20].

Until recently, there was no targeted therapy approved for the patients with advanced BTC who progressed on chemotherapy. Pemigatinib, a fibroblast growth factor receptor (*FGFR*) inhibitor, has recently been approved by the United States FDA in the second-line setting based on a phase II trial data (FIGHT-202; discussed in detail in Section 5) [21].

## 3. Genomic Profile of BTC

The advent of tumor genomic profiling by next-generation sequencing (NGS) techniques and increasing utilization of NGS in routine clinical practice have led to a vast repository of information on the genomic landscape of BTC [22,23,24,25,26] (Figure 1).

BTCs are genetically diverse cancers with remarkable variations in genetic alterations, although some specific patterns have been observed in a given anatomic subtype (Table 1).

In general, isocitrate dehydrogenase (IDH) and *FGFR* aberrations tend to cluster in iCCA, whereas *HER2 (ERBB2*) aberrations are more frequent in eCCA and GBC [28]. Javle et al. [23] reported comprehensive genomic profiling data on a large cohort of BTC patients that consisted of 412 patients with iCCA, 57 patients with eCCA, and 85 patients with GBC. This study also analyzed the correlation between clinical outcome and molecular profile in 321 patients receiving standard and experimental therapies. The study results showed marked variation in genetic alterations depending on the anatomic subtypes of BTC, with a predominance of *P53* (27%) in iCCA, *KRAS* (42%) in eCCA, and *ERBB2* (16%) in GBC. Furthermore, the study reported that *FGFR* (11%) and *IDH* mutations (20%) were mostly limited to iCCAs, and appeared to be mutually exclusive. In the iCCA group, *TP53* and *KRAS* mutations were associated with poor OS, whereas *FGFR2* aberrations were associated with improved OS. Another NGS-based study of tumor samples from 195 patients (78% iCCA and 22% eCCA) reported similar heterogeneity, with a preponderance of *IDH1* (30%), *ARID1A* (23%), *BAP1* (20%), *TP53* (20%), and *FGFR2* gene fusions (14%) in iCCA [25].

Accumulating data indicate that *IDH1/2* and other frequent mutations have distinct clinical and prognostic implications in large duct versus small duct subtypes of iCCA. IDH1/2 mutations portend a favorable prognosis in the small duct type of iCCA, but not in the large duct type, as reported in a study by Ma et al. [29] Furthermore, BAP1 expression loss correlated with favorable prognosis only in the large duct type of iCCA.

The genomic profile of BTC has a strong etiological correlation. Ong et al. [30] reported a significantly higher prevalence of *P53, SMAD4, MLL3*, and *GNAS* mutations in liver-fluke-associated CCAs, whereas non-liver-fluke-associated CCAs had higher rates of *IDH1/2* and *BAP1* mutations. Higher prevalences of *TP53* mutations in patients with hepatitis-B-associated iCCA and *KRAS* mutations in hepatitis-B-negative patients have been suggested in a study [31]. Furthermore, a recent international collaborative study reported four distinct molecular subtypes of CCA by performing an integrative clustering analysis of the combined clinical and genomic data of 489 patients from ten countries [24]. The CCA cluster associated with liver fluke infection (cluster 1 and 2) had reduced survival, was enriched with *ERBB2* amplification and *TP53* mutations, and had lower rates of *FGFR/IDH* alterations. The fluke-infection-negative (clusters 3 and 4) patients demonstrated high copy-number alterations and programmed cell death receptor ligand 1/programmed cell death receptor ligand 2 (PD-1/PD-L2) expression or epigenetic mutations (*IDH1/2* and *BAP1*), as well as *FGFR*- and *PRKA*-related gene rearrangements.

Genomic profile also varies considerably based on geographical location. A recently published study compared genomic profiling of 164 Chinese patients and 283 Western patients with iCCA [32]. The study reported a higher rate of mutations involving the DNA repair genes and higher tumor mutation burden (TMB) values (>10 mut/Mb) in the Chinese cohort, compared to the Western cohort. Conversely, the Western cohort had a higher prevalence of actionable driver mutations. These differences likely reflect the underlying etiological factors in different geographical regions.

## 4. Circulating Tumor DNA (ctDNA)-Based Molecular Profiling

Recent advancements of technologies have enabled molecular profiling of solid tumors, including BTCs, utilizing cell-free circulating tumor DNA (ctDNA) in the peripheral blood samples. ctDNA-based profiling is particularly useful in BTCs, given that tumor biopsy samples are often inadequate for sequencing. Tissue biopsies are often difficult to obtain, especially in extrahepatic and hilar CCA. In one meta-analysis of 16 studies, the diagnostic sensitivity of endoscopic retrograde cholangiopancreatography (ERCP)-guided brushings was approximately 42% [33]. Moreover, endoscopy-based procedures are associated with various risks inherent in invasive procedures, including acute pancreatitis, which occurs in approximately 2.6% of patients undergoing ERCP. Emerging data suggest that intratumor heterogeneity and clonal evolution are inherent to malignant tumors, often driven by systemic therapies [34]. Tissue biopsy may be of limited use due to intratumor heterogeneity, and repeated tissue biopsies to monitor clonal evolution are not feasible for most patients. Conversely, ctDNA analysis can be performed in peripheral blood samples obtained by outpatient phlebotomy, a minimally invasive procedure that can be repeated as needed. Consequently, ctDNA-based profiling of advanced BTC is gaining credence.

Early feasibility studies with ctDNA have reported high blood/tissue concordance rates of genomic alterations in patients with BTC, ranging from 74% to 100% (Table 2). In one study, the concordance was higher between ctDNA and metastatic site tissue-DNA than between ctDNA and primary tumor DNA (78% versus 65% for TP53, 100% versus 74% for KRAS, and 100% versus 87% for PIK3CA) [35]. In this study, 80 patients received molecularly matched therapy and had a significantly longer progression-free survival (HR, 0.60; 95% CI, 0.37–0.99; *p* = 0.047) than patients who received unmatched treatment. The study reported by Ettrich et al. showed that the mutational profile of the 23 available blood-tumor pairs was concordant in 74% of patients, with a higher concordance rate in iCCA (92%) versus eCCA (55%) [36]. These studies suggest that ctDNA-based molecular profiling is feasible and can be clinically useful in patients with BTC.

Furthermore, the feasibility of detecting the emergence of acquired resistance to targeted therapy by serial ctDNA analysis has also been reported in a study [34]. In this study, three patients with advanced iCCA enrolled in a phase II study of an FGFR inhibitor (BGJ398) underwent serial ctDNA testing at enrollment and after radiologic progression. ctDNA analysis demonstrated the appearance of de novo point mutations in the *FGFR2* gene, conferring resistance to BGJ298.

The largest study that evaluated the utility of ctDNA analysis in patients with advanced BTC enrolled 124 patients (70% patients with iCCA) who underwent ctDNA-based molecular profiling utilizing a clinically available NGS-based assay (Guardant 360) [39]. In this study, at least one therapeutically relevant alteration was observed in 55% of patients, and 21% of patients had one of the frequently occurring alterations, which included *BRAF* mutations, *ERBB2* amplification, *FGFR2* fusions, *FGFR2* mutations, and *IDH1* mutations. On average, each sample contained three alterations, with a median allelic fraction of 0.52%. Moreover, this study demonstrated that the overall landscape of alterations was different in iCCA versus eCCA/GBC. Therapeutically relevant alterations were more frequent in iCCA compared to eCCA/GBC, with the *FGFR2* alteration being the most frequent (7%) in patients with iCCA. *ERBB2* alteration was exclusively found in eCCA. The ARID1A alteration was more common in eCCA, and *CDK6*, *APC*, and *SMAD4* alterations were more common in GBC. This study did not report the proportion of patients receiving ctDNA-guided targeted therapies or their outcomes.

Although promising, ct-DNA based profiling of BTC has several limitations related to the technology itself and its applicability. Wide variations exist in the preanalytical variables, assay characteristics (PCR-based versus NGS-based techniques), bioinformatic analysis methods, sensitivity, and specificity [40]. These factors pose significant challenges in the interpretation of clinical trial data utilizing different ctDNA platforms. Furthermore, robust prospective clinical trial data are needed to validate the clinical utility of ctDNA in the management of advanced BTC before ctDNA-based profiling can be widely accepted in routine clinical practice.

## 5. Targeted Therapies

Studies utilizing NGS to elucidate the molecular profile of BTC have highlighted that BTCs are target-rich malignancies with therapeutically relevant genetic alterations identified in approximately half of the patients [25,26], which has led to a plethora of clinical trials with targeted therapies. Although no randomized study reported to date has confirmed the superiority of the targeted approach in advanced BTC, the MOSCATO-01 trial provides early supportive evidence [41]. In this trial, patients who received targeted therapy (18 out of 43 patients) had an improved median OS compared to those treated with unselected therapies (median OS 17 vs. 5 months; *p* = 0.008).

Currently, the most promising targets in advanced BTCs consist of *IDH* mutations, *FGFR* aberrations, *BRAF* mutations, the DNA damage repair (DDR) pathway, and the *HER2* pathway. Table 3 summarizes the published study results of the targeted therapies in advanced BTC.

### 5.1. FGFR Pathway

Targeting *FGFR* fusions, almost exclusively seen in iCCA, has shown clinically meaningful benefits in recent clinical trials. *FGFR* gene fusions are present in approximately 20% of patients with iCCAs (Table 1). *FGFR*s consist of four transmembrane receptors (*FGFR* 1–4) with intracellular tyrosine kinase domains that regulate cell proliferation, migration, differentiation, and angiogenesis via binding to the ligands (fibroblast growth factors, FGFs) [54]. *FGFR2* fusion is the most common *FGFR* aberration, which activates multiple oncogenic canonical signaling events downstream of *FGFR* [55]. FGFR2 fusions are therapeutically more relevant than *FGFR* mutations [21].

Among the *FGFR* tyrosine kinase inhibitors, Pemigatinib (INCB054828), Infigratinib (BGJ398), and Futibatinib (TAS-120) have shown encouraging results in phase I and II studies with manageable toxicity profiles (Table 3). These selective *FGFR* inhibitors are associated with an ORR of 15–35%, and a median PFS of around six months in previously treated patients with *FGFR2* fusions. Pemigatinib, a selective oral inhibitor of *FGFR1-3*, has been evaluated in a phase II study (FIGHT-202) that enrolled chemotherapy-refractory patients with a variety of *FGFR* alterations [21]. This study reported an ORR of 35.5%, a median PFS of 6.9 months, and a preliminary OS of 21.1 months in patients with *FGFR2* fusions. This study result led to the FDA approval of pemigatinib for the treatment of chemotherapy-refractory advanced iCCA, the first approval of a targeted agent for advanced BTC. Of note, pemigatinib had no significant activity in patients with *FGFR* amplification or mutations (ORR 0% and a median PFS of 2.1 months). Adverse events were manageable, with hyperphosphatemia being the most common any-grade adverse event (60%). Grade ≥ 3 adverse events were reported in 64% of patients, which included hypophosphatemia (12%), arthralgia (6%), stomatitis (5%), hyponatremia (5%), abdominal pain (5%), and fatigue (5%). Serious adverse events occurred in 45% of patients, but no treatment-related deaths were reported. Hyperphosphataemia appears to be related to the alteration of vitamin D and phosphorus metabolism mediated via fibroblast growth factor 23 (FGF23) [56]. Currently, Pemigatinib, infigratinib, and futibatinib are being investigated in treatment-naive patient cohorts in phase III trials against the gemcitabine and cisplatin combination (Table 4). ARQ087 (derazantinib), a nonselective multikinase inhibitor with activity on *FGFR*, has also recently entered phase III trial in pretreated patients after the publication of encouraging phase II data [45]. The study results of other *FGFR* inhibitors, including Debio 1347 and Erdafitinib, are summarized in Table 3.

*FGFR* inhibitors are subject to a variety of resistance mechanisms, including polyclonal point mutation in the *FGFR2* kinase domain [38] and the development of novel *FGFR2* fusions [57]. Futibatinib (TAS-120), a highly selective pan-*FGFR* inhibitor, has activity against *FGFR2* resistance mutations [43] and has shown promising clinical activity in *FGFR*-aberrant iCCA patients in an early-phase study, including in patients progressing on prior *FGFR* inhibitors. The phase II FOENIX-101 study with futibatinib is currently recruiting patients with iCCA-harboring *FGFR2* gene rearrangements after progression on first-line treatment (Table 4).

### 5.2. Isocitrate Dehydrogenase (IDH) Mutations

Approximately 20% of patients with iCCA harbor *IDH1* or *IDH2* mutations (Table 1). *IDH1* mutations are more frequent than *IDH2* mutations and are generally found in iCCAs not related to hepatitis and fluke infection [58]. These somatic mutations cause an increase in *IDH1/2* activity resulting in modulation of cell metabolism and accumulation of 2-hydroxyglutarate (2-HG), an oncometabolite that interferes with normal cell differentiation and promotes tumorigenesis [59]. *IDH1* and *IDH2* mutations do not have any prognostic implications, unlike *FGFR* alterations [60].

Ivosidenib (AG-120), an oral *IDH1* inhibitor, has shown encouraging activity in patients with advanced *IDH1*-mutant, chemotherapy-refractory CCA [48,49] (Table 3). The phase I study [48] with ivosidenib in refractory patients reported an ORR of 5%, a median PFS of 3.8 months, and a median OS of 13.8 months. Although the response rate was low, a significant proportion of patients were progression-free at 12 months (21.8%), and the tolerability of ivosidenib was remarkable—only 5% of patients experienced grade 3 or higher toxicities, and there were no dose-limiting toxicities. The most frequent side effects, including fatigue and nausea, were manageable. Subsequently, a phase III, placebo-controlled trial (ClarIDHy) was conducted in patients with *IDH1*-mutant advanced iCCA who progressed on at least one line of chemotherapy [49]. The primary endpoint was PFS, and crossover from placebo to ivosidenib was permitted after progression. This study demonstrated a small median PFS improvement with ivosidenib over placebo, 2.7 months vs. 1.4 months (HR 0.37; 95% CI, 0.25–0.54; *p* < 0.0001). Although the short PFS improvement is a cause for concern, 32% of patients on the ivosidenib treatment arm had not progressed at six months, and 22% had not progressed at one year. Conversely, no patient on the placebo arm was progression-free at six months. The median OS in the ivosidenib arm was 10.8 months vs. 9.7 months in the placebo arm, which was not statistically significant. However, when adjusted for the crossover, the median OS in the placebo arm dropped to six months, and the OS difference was significant (HR 0.46; *p* = 0.0008). The toxicity profile was consistent with the previous report [48].

Given the modest activity of *IDH* inhibitors, novel strategies targeting *IDH* mutations are being explored in clinical trials. iCCA cell lines showed remarkable sensitivity to dasatinib, a multikinase inhibitor, in a preclinical study utilizing a high-throughput drug screening method, which was confirmed in a xenograft model [61]. Currently, dasatinib is being investigated in a phase II trial in patients of advanced iCCA harboring *IDH*-mutations (ClinicalTrials.gov Identifier: NCT02428855). Another preclinical study demonstrated that 2-HG produced as a result of *IDH* mutations suppresses homologous recombination and induces PARP (poly ADP ribose polymerase) inhibitor sensitivity (‘BRCAness’) [62], which led to a phase II trial with Olaparib in solid tumors, including CCA (ClinicalTrials.gov Identifier: NCT03212274). Furthermore, a phase I study is planned with a combination of ivosidenib and cisplatin/gemcitabine in patients with advanced CCA (ClinicalTrials.gov Identifier: NCT04088188).

### 5.3. Human Epidermal Growth Factor Receptor (HER) Pathway

The HER family receptors consist of four distinct receptors: epidermal growth factor receptor (EGFR) or HER1, HER-2, HER-3, and HER-4. In normal cells, binding of the ligands to the extracellular domain of these receptors leads to the dimerization of the receptors with eventual phosphorylation of the intracellular tyrosine kinase domain and activation of the downstream pathways, which include MAPK, PI3K/AKT/mTOR, and STAT pathways [63].

Therapies targeting HER2 have demonstrated modest activity thus far. Among the 5–15% of BTCs expressing HER2, most are gallbladder cancers or eCCA [64]. The combination of HER2 directed monoclonal antibodies pertuzumab and trastuzumab was investigated as a part of the MyPathway multi-basket study in 11 patients with refractory BTC [65]. At a median follow-up of 4.2 months, four patients had partial responses (PR), and three had stable disease (SD) for about four months. Neratinib, an oral tyrosine kinase inhibitor of *EGFR*, *HER2*, and *HER4*, was studied in *HER2*-mutant advanced solid tumors in the phase II SUMMIT basket trial [51]. An ORR of 10% was reported among the 20 patients with HER2-mutant BTC. Trastuzumab deruxtecan (DS-8201), an antibody-drug conjugate targeting HER2, is under evaluation in HER2-positive patients with advanced BTC in a clinical trial (Table 4).

EGFR (HER1) overexpression is common among among patients with CCA and is associated with poor prognosis, particularly in iCCA [66]. A phase III study investigated the activity of an EGFR inhibitor, erlotinib, plus chemotherapy (gemcitabine and oxaliplatin) versus chemotherapy alone in therapy-naive patients with advanced BTC [67]. The combination of erlotinib and chemotherapy was associated with a significantly increased ORR (30% vs. 16%; *p* = 0.005), but an improvement in PFS or OS was not demonstrated. A post-hoc analysis of this study showed that the subgroup of patients with CCA had an improved median PFS with the combination therapy (5.9 months vs. 3.0 months; *p* = 0.049). Studies with a combination of chemotherapy and anti-EGFR antibodies cetuximab [68] and panitumumab [69] failed to show an improvement in ORR, PFS, or OS. Studies with other HER pathway inhibitors, including afatinib [70] and Lapatinib [71], also showed disappointing results. Conversely, varlitinib, a pan-*HER* inhibitor, demonstrated a somewhat encouraging result, with a partial response rate of 27%, stable disease of 43%, and a disease control rate of 70% in 37 patients with refractory BTC in a pooled analysis of three phase I studies [72]. Currently, varlitinib, in combination with gemcitabine and cisplatin, is being evaluated in patients with treatment-naive advanced BTC (ClinicalTrials.gov Identifier: NCT02992340) (Table 4).

### 5.4. DNA Damage Repair (DDR) Mechanisms and BAP1 Mutations

Somatic or germline mutations in *BRCA* are detected in about 3.5% of patients with BTC and result in an immunogenic tumor profile charact-herized by a higher TMB and a higher rate of microsatellite instability (MSI)-high tumors [73]. Furthermore, *IDH* mutations confer sensitivity to PARP inhibitors, as discussed above. DDR pathway mutations or *IDH1* mutations were reported to be present in about half of patients with BTC [74]. Based on this rationale, a phase II study is underway to test the combination of rucaparib (a PARP inhibitor) and nivolumab (a PD-1 blocker) in a cohort of patients who progressed on chemotherapy (Table 4). Patients of CCA (mostly iCCA) with mutations in *BAP1*, a tumor suppressor gene involved in DNA double-strand break repair, are reported to have an aggressive disease and poor response to standard therapies [75]. A phase II basket trial is currently investigating the clinical benefit of the PARP inhibitor niraparib in patients with *BAP1* mutations and other DDR-deficient solid tumors, including CCA (Table 4).

### 5.5. RAS/RAF/MEK/ERK Pathway

Although *RAS* mutations serve as oncogenic drivers in many different types of malignancies, targeting *RAS* has mostly been unsuccessful because of its intricate interactions with other signaling pathways. Consequently, inhibiting targets downstream of *RAS*, such as *BRAF* and *MEK*, has been attempted, but with limited clinical success thus far. Selumetinib, a *MEK* inhibitor, in combination with cisplatin/gemcitabine has been studied in a phase Ib study (ABC-04) in therapy-naive patients with advanced BTC [76]. Among eight evaluable patients, three had a partial response, and five had stable disease with a median PFS of 6.4 months. A phase II study with selumetinib, which included 39% of chemotherapy-refractory patients, showed an ORR of 12%, a median PFS of 3.7 months, and a median OS of 9.8 months with an acceptable safety profile [77].

*BRAF* mutations have been described in less than 5% of patients with BTC, primarily in the cohort with intrahepatic disease [23]. Dual targeting with dabrafenib (*BRAF* inhibitor) and trametinib (*MEK* inhibitor) within the ROAR trial, a phase II basket trial of 178 patients harboring the *BRAF^V600E^* mutations, which included 33 patients with advanced refractory CCA, showed encouraging efficacy with an ORR of 41%, a median PFS of 7.2 months, and a median OS of 11.3 months [50]. PLX8394, a BRAF inhibitor, is currently being evaluated in patients with refractory solid tumors, including BTC (ClinicalTrials.gov Identifier: NCT02428712).

### 5.6. PI3K/AKT/mTOR Pathway

The PI3K/AKT/mTOR pathway regulates cellular proliferation and angiogenesis through its interactions with the RAS/RAF/MEK pathway and the mTOR signaling pathway [78] (Figure 1). Mutations in *PIK3CA*, resulting in upregulation of the PI3K/AKT/mTOR pathway, have been identified in many cancer types, including in BTC [79]. Everolimus, an mTOR inhibitor, has shown modest activity in patients with previously treated BTC in a phase II trial, which reported a median PFS of 3.2 months and a median OS of 7.7 months [80]. A phase II study with everolimus in the first-line setting demonstrated a PFS of 5.5 months and OS of 9.5 months [81]. A combination of cisplatin/gemcitabine and copanlisib, a pan-*PI3K* inhibitor, has shown modest activity in a phase I study with an ORR of 17.4% [82]. A phase II study with this combination in the first-line setting is planned (Table 4).

### 5.7. Vascular Endothelial Growth Factor (VEGF)

Vascular endothelial growth factor (VEGF) overexpression in BTC is a poor prognostic factor, particularly in iCCA [66]. However, targeting VEGF has not been a successful strategy so far. A meta-analysis of seven randomized trials involving 964 patients reported a lack of PFS or OS benefit with anti-VEGF therapy in combination with chemotherapy compared to chemotherapy alone [83]. Cediranib, a potent VEGF receptor tyrosine kinase inhibitor, did not show a survival benefit in therapy-naive patients when combined with gemcitabine/cisplatin in comparison to gemcitabine/cisplatin plus placebo in the randomized phase II ABC-03 trial [84]. Studies with other multikinase VEGF receptor inhibitors, including sorafenib [85], lenvatinib [86], vandetinib [87], and regorafenib [88], have also been disappointing. Alternative strategies, for example, a combination of VEGF inhibition (lenvatinib) and immune checkpoint inhibition (pembrolizumab), have reported encouraging results (detailed in Section 5.9).

### 5.8. Miscellaneous Pathways

*NTRK* (neurotropic tyrosine kinase receptor) fusions, identified in 3.5% of patients with iCCA [89], are targetable with currently approved first-generation TRK (tropomyosin receptor kinase) inhibitors larotrectinib and entrectinib, which have produced an impressive ORR of 57% to 75% in advanced solid tumors harboring *NTRK* fusions [90,91]. Of note, TRK inhibitors have activity against *ROS1* and *ALK* fusions, which are reported to be present in 0–8.7% and 2.7% of patients with CCA, respectively [89,92].

*c-MET* regulates cell proliferation, migration, and invasion [93]. A phase II study with cabozantinib, an oral *MET* inhibitor, reported a median PFS of 1.8 months and a median OS of 5.2 months in patients with previously-treated advanced CCA [94]. A phase I study of an oral *c-MET* inhibitor, tivantinib, in combination with gemcitabine, demonstrated a modest activity with an ORR of 19% [95]. Merestinib, a small molecule inhibitor of *MET*, did not improve ORR, PFS, or OS when added to gemcitabine and cisplatin in a recently reported randomized phase II trial [96].

### 5.9. Targeted Therapy and Immunotherapy Combinations

Although a detailed account of immunotherapy in BTC is beyond the scope of this article, a discussion of novel immunotherapy/targeted therapy combinations is relevant. The modest activity of single-agent immunotherapeutic agents in advanced BTC, reporting an ORR of 5–20% [97], led to the exploration of a variety of combination strategies, including immunotherapy/targeted therapy combinations. The combination of pembrolizumab (a PD-1 inhibitor) and ramucirumab (a VEGF receptor 2 inhibitor) investigated in a phase I trial in patients with previously treated advanced BTC reported a modest median PFS and median OS of 1.6 and 6.4 months, respectively [98]. A promising result has been reported with the combination of a multikinase inhibitor, lenvatinib, and pembrolizumab in 32 patients who had received at least two prior anticancer therapies [99]. This study reported an ORR of 25%, a median PFS of 4.9 months, and a median OS of 11 months. A multicenter randomized phase II trial (*n* = 86) of atezolizumab (a PD-L1 blocker) as monotherapy or in combination with cobimetinib (a MEK inhibitor) in refractory advanced BTC has recently reported superior median PFS with the atezolizumab/cobimetinib combination—3.65 months vs. 1.87 months (*p* = 0.027); OS data are not mature at this time [100]. A combination of PD-1 inhibitor and DNA repair modulators, as described in Section 5.4, is another area of exploration. These early results suggest that the immunotherapy/targeted therapy combinations will likely be a new frontier for further exploration.

## 6. Future Directions and Conclusions

Advanced BTC is a malignancy associated with poor prognosis and rising incidence worldwide. However, evolving targeted therapies have instilled new hope in the treatment paradigm of BTC. The advancement of genomic profiling technologies has helped unravel the molecular heterogeneity of BTC and numerous targetable molecular alterations that have encouraged a concerted move towards the adoption of a ‘precision medicine’ approach. The clinical trials with targeted therapies have already produced early promising results in a chemotherapy-refractory setting. Consequently, all patients with advanced BTC must have access to tumor molecular profiling at the time of diagnosis.

To further support the precision oncology approach, several expert-curated knowledge bases have emerged [101], which annotate published information on the clinical implication of specific molecular alterations. However, a universal definition of ‘actionable alteration’ is still lacking. Furthermore, accumulating data indicate that significant genomic heterogeneity exists in a given tumor sample [102], making target selection difficult. Consequently, it is not surprising that the targeted approaches against BTC have not produced robust responses thus far. These practical hurdles will need to be overcome to exploit the knowledge of precision medicine fully.

All patients on targeted therapies eventually develop treatment resistance, and repeat genomic profiling at the time of progression may be valueable. Repeat profiling might help elucidate the resistance mechanisms and reveal a new target. However, a tumor tissue sample for genomic profiling is not available in a large number of patients, which underscores the importance of exploring ctDNA-based genomic profiling. As discussed earlier, ctDNA-based profiling has a high concordance rate with tissue-based profiling [35,36]. As ctDNA-based genomic profiling becomes standardized and readily available, patient enrollment in clinical trials based on ctDNA profiling will help further advancement in this field. Furthermore, ctDNA-based testing might be of particular value for serial assessments of tumor genomic profiles.

Although targeted therapies in BTC are promising, a large group of patients do not harbor a known targetable mutation. Therefore, it is imperative that further exploration should be made to gain a deeper understanding of the complex crosstalk among the oncogenic signaling pathways and to identify additional actionable targets. It is well-known that tumor growth is dependent on multiple signaling pathways, which has prompted many ongoing clinical trials to evaluate combination therapies targeting multiple molecular alterations simultaneously that may improve response rates and durability of the responses. We anticipate that these trial results will help to refine future therapies and drug development for patients with advanced BTC.

## Figures and Tables

**Figure 1 cancers-12-02039-f001:**
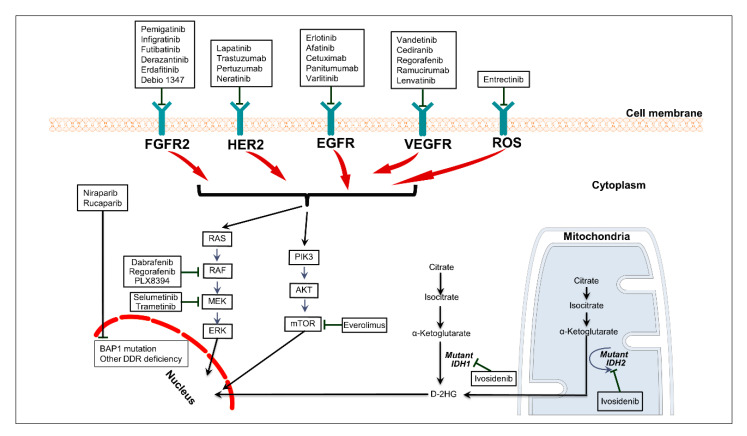
Interactions among the therapeutically relevant signaling pathways in biliary tract cancer and therapies available to target those pathways. Abbreviations: DDR, DNA damage repair; 2-HG, 2-hydroxyglutarate.

**Table 1 cancers-12-02039-t001:** Prevalence of therapeutically relevant genetic alterations in patients with advanced biliary tract cancer and the corresponding targeted therapies [27].

Genetic Alteration	Extrahepatic Cholangiocarcinoma (%)	Intrahepatic Cholangiocarcinoma (%)	Gallbladder Cancer (%)	Targeted Therapy
*FGFR*	1	20	3	Pemigatinib, infigratinib, futibatinib, derazantinib, erdafitinib
*IDH 1 and 2*	3	20	2	Ivosidenib, AG-881, FT 2102, BAY1436032
*BRAF*	3	3	4	Dabrafenib, PLX 8394
*ERBB2/3*	15	7	20	Neratinib, trastuzumab, varlitinib
*cMET*	3	5	0	Cabozantinib, tivantinib, merestinib
*PIK3CA*	7	6	10	Copanlisib
*NTRK*	4	4	4	Larotrectinib, entrectinib
*BAP1*	1	13	0	Niraparib

**Table 2 cancers-12-02039-t002:** Studies evaluating the clinical utility of circulating tumor DNA (ctDNA) in patients with advanced biliary tract cancer (BTC).

Study	Patient Population	*n*	ctDNA Assay	Study Design	Mutations Tested in the Plasma	Major Findings
Andersen et al. [37]	CCA	5	Multiplex digital PCR	Plasma samples from CCA patients with known tumor mutations were tested to determine blood/tissue concordance	Total 31 mutations in the *KRAS, NRAS, BRAF*, and *PIK3CA* genes.	All known mutations in the tumor were detected in plasma samples in all patients.
Ettrich et al. [36]	CCA (54% iCCA, 46% eCCA)	24	NGS-based assay	Tumor tissue and corresponding plasma samples were collected to investigate blood/tissue concordance	15 genes frequently mutated in CCA *	The blood/tissue concordance was 74% overall and 92% for intrahepatic CCA. The VAF in ctDNA correlated with the tumor burden, and in iCCA, with the PFS.
Goyal et al. [38]	Phase I study of BGJ398 in iCCA patients	3	Droplet digital PCR	ctDNA analyses on serial plasma samples before and after treatment to determine the resistance mechanism	*FGFR2* mutations	In all 3 cases, post-progression sequencing of the *FGFR2* gene demonstrated de novo point mutations that conferred resistance to BGJ298
Mody et al. [39]	Advanced BTC (70% iCCA)	124	NGS-based assay developed by Guardant Health	Plasma samples obtained to find therapeutically relevant alterations as part of routine clinical care	73-gene panel	Therapeutically relevant alterations were observed in 55% of patients (21% of patients had one of the following alterations—*BRAF* mutations, *ERBB2* amplification, *FGFR2* fusions, *FGFR2* mutations, and *IDH1* mutations).

Abbreviations: CCA, cholangiocarcinoma; PCR, polymerase chain reaction; iCCA, intrahepatic cholangiocarcinoma; eCCA, extrahepatic cholangiocarcinoma; VAF, variant allele fraction; PFS, progression-free survival; NGS, next-generation sequencing.* *TP53*, *KRAS*, *ARID1A*, *BAP1*, *PBRM1*, *PIK3CA*, *SMAD4*, *FBXW7*, *IDH1*, *BCL2*, *BRAF*, *CDKN2A*, *ERBB2*, *IDH2*, and *NRAS*.

**Table 3 cancers-12-02039-t003:** Published study results of targeted therapies in patients with advanced biliary tract carcinoma (BTC).

Study Drug	*n*	Study Phase	Study Population	Pathway Targeted	ORR (%)	Median PFS (Months)	Median OS (Months)	Comments
Multiple targeted agents (MOSCATO-01) [41]	43	Prospective, molecular triage trial	Refractory BTC	-	33	5.2	17	18 of 43 patients received targeted agents resulting in improved median OS of 17 months vs. 5 months in patients who did not have an actionable target. Targetable alterations present: *IDH1/2* (18%), *FGFR1/2* (16%), *EGFR, ERBB2 or ERBB3* (16%), *PTEN* (14%), *MDM2* (10%) and *PIK3CA* (10%).
Pemigatinib [21]	107	II (FIGHT-202)	Refractory iCCA	*FGFR*	35.5	6.9	21.1	Recently approved by FDA in the second-line setting
Infigratinib [42] (BGJ398)	61	II	Refractory iCCA	*FGFR*	14.8	5.8	-	Disease control rate was 75.4%
Futibatinib [43] (TAS-120)	45	I	Refractory iCCA including patients who failed other FGFR inhibitor	*FGFR*	25	-	-	Median duration of treatment 7.4 months
Futibatinib [44]	67	II (FOENIX-CCA2)	iCCA	*FGFR*	34.3	-	-	Median duration of response was 6.2 months (range, 2.1–14.2).
Derazantinib (ARQ 087) [45]	29	I/II	Refractory iCCA	*FGFR*	20.7	5.7	-	The disease control rate was 82.8%. OS not reached after a median follow up of 20 months
Erdafitinib [46]	17	II	Refractory iCCA	*FGFR*	47	5.6	-	In patients with *FRFR2/3* fusion: ORR 67%, median PFS 12.6 months
Debio 1347 [47]	9	I	Refractory solid tumors	*FGFR*	22	-	-	The median time on treatment was 24 weeks (range, 4–57 weeks)
Ivosidenib [48]	73	I	Refractory iCCA	*IDH1*	5	3.8	13.8	MTD was not reached; 500 mg daily was selected for expansion.
Ivosidenib (AG-120) [49] vs. placebo	185	III (ClarIDHy)	Refractory iCCA	*IDH1*	2.4	2.7 vs. 1.4, *p* < 0.001	10.8 vs. 9.7, *p* = 0.06	PFS rates at 6 and 12 months were 32.0% and 21.9% in ivosidenib arm. Grade 3 adverse events: 46% in ivosidenib arm vs. 36% with placebo.
Dabrafenib + Trametinib [50]	33	II (ROAR)	Refractory solid tumors including CCA	Combined *BRAF + MEK* inhibition	41	7.2	11.3	7 of 13 responders (54%) had a duration of response ≥ 6 months.
Neratinib [51]	20	II (SUMMIT)	Refractory BTC	*EGFR, HER2*, and *HER4*	10	1.8	-	Most common grade 3 or 4 adverse event was diarrhea (20%)
Regorafenib [52]	68	Randomized phase II (REACHIN)	Refractory BTC	VEGF	0	3 vs. 1.5, *p* = 0.004	5.3 vs. 5.1, *p* = 0.21	Median treatment duration is 10.9 weeks for regorafenib vs. 6.3 weeks for placebo (*p* = 0.004).
Varlitinib plus capecitabine vs. placebo + capecitabine [53]	127	Randomized phase II (TreeTopp)	Refractory BTC	*EGFR, HER2*, and *HER4*	9.4 vs. 4.8 (*p* = 0.42)	2.8 vs. 2.8	7.8 vs. 7.5	Exploratory analyses suggested that female patients and patients with gall bladder cancer achieved comparatively higher median PFS with the study agent

Abbreviations: ORR, overall response rate; PFS, progression-free survival; OS, overall survival; CCA, cholangiocarcinoma; iCCA, intrahepatic cholangiocarcinoma;BTC, biliary tract cancer.

**Table 4 cancers-12-02039-t004:** Selected ongoing and upcoming clinical trials evaluating targeted therapies in patients with advanced biliary tract cancer *.

Clinical Trials	Targeted Agent	Study Phase	Signaling Pathway Targeted	Primary Endpoint	Study Identifier (ClinicalTrials.gov Identifier)
First Line	Pemigatinib vs. gemcitabine+cisplatin	III	*FGFR*	PFS	NCT03656536 (FIGHT-302)
	Infigratinib vs. gemcitabine+cisplatin	III	*FGFR*	PFS	NCT03773302 (PROOF)
	Futibatinib vs. gemcitabine+cisplatin	III	*FGFR*	PFS	NCT04093362 (FOENIX)
	Varlitinib + gemcitabine + cisplatin	IB/II	EGFR, HER2, HER4	MTD, PFS	NCT02992340
	Copanlisib + gemcitabine + cisplatin	II	mTOR	PFS	NCT02631590
	Olaparib	II	DNA damage repair	ORR	NCT04042831
Second or Later Line	Derazantinib	II	*FGFR*	ORR, and PFS at 3 months	NCT03230318 (FIDES-01)
	BGJ398	II	*FGFR*	ORR	NCT02150967
	Futibatinib	II	*FGFR*	ORR	NCT02052778
	JNJ-42756493 (Erdafitinib)	II	*FGFR*	ORR	NCT02699606
	Ivosidenib or pemigatinib in combination with Gemcitabine and cisplatin	I	*FGFR* and *IDH1*	Tolerability	NCT04088188
	Rucaparib + nivolumab	II	*PARP* + PD-1	PFS rate at 4 months	NCT03639935
	Olaparib	II	*IDH* mutation associated ‘BRCAness’	ORR	NCT03212274
	Niraparib	II	*BAP1* and DDR pathway	ORR	NCT03207347
	Entrectinib	II	*NTRK, ROS1, or ALK*	ORR	NCT02568267
	PLX8394	I/II	*BRAF*	Pharmacokinetics, ORR	NCT02428712
	Gemcitabine + selumetinib vs. gemcitabine	II	*MEK*	ORR	NCT02151084
	AG-881	I	*IDH*	Safety, MTD, RP2D	NCT02481154
	FT 2102	Ib/II	*IDH1*	MTD, RP2D, ORR	NCT03684811
	BAY1436032	I	*IDH1*	Safety, MTD, RP2D	NCT02746081
	Afatinib + capecitabine	I	*EGFR*	Safety, MTD, RP2D	NCT02451553
	Trastuzumab deruxtecan (DS-8201)	II	*HER2*	ORR	JMA-IIA00423 (HERB)
	KA2507	Ib/II	HDAC	MTD, PFS rate at 4 months	NCT04186156
	Fruquintinib	II	VEGFR	PFS	NCT04156958

Abbreviations: PFS, progression-free survival; MTD, maximal tolerated dose; ORR, overall response rate; RP2D, recommended phase II dose. * Information obtained from ClinicalTrials.gov (https://clinicaltrials.gov/), accessed between 17 May 2020, and 21 June 2020.

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
