# Peer review of "Targeted Therapies in Advanced Biliary Tract Cancer: An Evolving Paradigm"

_cancers, 2020, doi:10.3390/cancers12082039_

Round 1

Reviewer 1 Report

This is a timely, topical review of molecular targeted therapies of BTCs. They also describe the usefulness of ctDNA, but do not mention the potential for combination therapy with immuno-oncology drugs.

Since ctDNA-based profiling becomes popular, can it be described in the abstract?

Again, they may mention immuno-oncology in the discussion part. Maybe focus on the combination with molecular targeted drugs.

Author Response

Dear Reviewer 1,

We greatly appreciate the review and the comments. We would like to express our most sincere gratitude for the thorough review and the suggestions. We have made a sincere effort to address your critique and suggestions in our revised manuscript, which, we believe, will enhance the value of this article. Our revised manuscript has been submitted with all modifications shown as tracked changes.

Below is our point by point response to the critique-

  1. This is a timely, topical review of molecular targeted therapies of BTCs. They also describe the usefulness of ctDNA, but do not mention the potential for combination therapy with immuno-oncology drugs.

A new paragraph highlighting the study results of immunotherapy/targeted therapy combinations in advanced BTC has been added (section 5.9) as follows-

Although a detailed account of immunotherapy in BTC is beyond the scope of this article, a discussion of novel immunotherapy/targeted therapy combinations is relevant. The modest activity of single-agent immunotherapeutic agents in advanced BTC, reporting an ORR of 5-20%[102], led to the exploration of a variety of combination strategies, including immunotherapy/targeted therapy combinations. The combination of pembrolizumab (a PD-1 inhibitor) and ramucirumab ( a VEGF receptor 2 inhibitor) investigated in a phase I trial in patients with previously treated advanced BTC reported a modest median PFS and OS of 1.6 and 6.4 months, respectively[103]. A promising result has been reported with the combination of a multikinase inhibitor, lenvatinib, and pembrolizumab in 32 patients who had received at least 2 prior anticancer therapies[104]. This study reported an ORR of 25%, a median PFS of 4.9 months, and a  median OS of 11 months. A multicenter randomized phase 2 trial (n=86) of atezolizumab (a PD-L1 blocker) as monotherapy or in combination with cobimetinib (a MEK inhibitor) in refractory advanced BTC has recently reported superior median PFS with the atezolizumab/cobimetinib combination- 3.65 months vs. 1.87 months (p=0.027); OS data are not mature at this time[105]. A combination of PD-1 inhibitor and DNA repair modulators, as described in section 5.4, is another area of exploration. These early results suggest that the immunotherapy/targeted therapy combinations will likely be a new frontier for further exploration.

  1. Since ctDNA-based profiling becomes popular, can it be described in the abstract?

We appreciate this suggestion. We included the following sentence in the abstract highlighting the potential of ctDNA in the treatment of BTC.

‘Genomic profiling of cell-free circulating tumor DNA that can assist in the identification of an actionable target is another exciting area of development.’

  1. Again, they may mention immuno-oncology in the discussion part. Maybe focus on the combination with molecular targeted drugs.

It is addressed above (point 1).

We look forward to receiving your response.

Best regards,

Amit Mahipal

Reviewer 2 Report

The authors provide a well-writen state of the art review of conventional and targeted therapy in biliary tract cancer.

Author Response

Dear Reviewer 2,

We greatly appreciate the review. We also appreciate your kind comment recognizing our effort. Our revised manuscript has been submitted with all modifications shown as tracked changes.

We look forward to receiving your response.

Best regards,

Amit Mahipal

Reviewer 3 Report

  • It is a comprehensive and well written narrative review. In my opinion, the clinical/practical limitations of detecting targetable mutations and applicability should be adressed in more detail in the text.  

  • A narrative review may be a disadvantage in the field of biomarkers such as ctDNA (ctDNA is also discussed in in the text) due to potential heterogeneity of cohorts, use (or not) of adequate controls, quality material used, different techniques/platsforms used (with varying sensitivity/specificity) etc. A comment on these limitations in the text would be advised. 

  • The term ‘advanced BTC’ may implicate a very different tumor load and well known variety in mutational status, when referring to iCCA, pCCA or dCCA. Although the different anatomical subtypes are mentioned in the text (Table 1), the structure of text of this review focusses on BTC in general. I would suggest to try to make a distinction (when available) between the literature on iCCA (probably larger tumors) and a separate description of pCCA/dCCA (often smaller) and different mutations.

  • Within the intrahepatic CCA, we distinguish small duct and large duct intrahepatic cholangiocarcinomas  with different mutations described within these subtypes. Is it possible to discuss in more detail these subtypes and/or the (ir)relevance of morphology versus mutational status in the text?

  • In line 131-132 is it stated that ctDNA-based profiling is particularly useful in BTCs, given that tumor biopsy samples are often inadequate for sequencing. The authors may consider to adress this topic (problems with biopsy) in more detail. It is a well known problem in CCA that the tumor tissue/tumor cells maybe difficult to obtain, even in the non-curative setting. What about the clinical feasibility to detect targetable mutations on biopsy/brush? And is tumor heterogeneity a relevant issue in tissue?

  • Line 143-147 descibes ctDNA in 3 patients with FGFR mutations and ‘advanced BTC’. These tumors are probably iCCA.   What about therapeutic monitoring with ctDNA in the context of pCCA/dCCA?

From line 147-154 the largest study is described with 70% iCCA: was it possible to separately evaluate value of ctDNA in the 70% of iCCA cases versus other localisations?

  1. How strong is the evidence for value of ctDNA in iCCA versus other anatomical localisations for instance?

  • Line 349-350: “All patients on targeted therapies eventually develop treatment resistance, which calls for a repeat genomic profiling at the time of progression.” Does this strong statement follow from this review? Is it validated for BTC, including value of repeat genomic profiling ( a few exemples are mentioned in the text, such as FGFR)?     

Author Response

Dear Reviewer 3,

We greatly appreciate the review and comments. We would like to express our most sincere gratitude for the thorough review and the suggestions. We have made a sincere effort to address your critique and suggestions in our revised manuscript, which, we believe, will enhance the value of this article. Our revised manuscript has been submitted with all modifications shown as tracked changes.

Below is our point by point response to the critique-

  1. It is a comprehensive and well written narrative review.

We appreciate your kind comment recognizing our effort.

  1. In my opinion, the clinical/practical limitations of detecting targetable mutations and applicability should be addressed in more detail in the text.

To address this critique, we have added the following text in section 6 (line 396-402):

To further support the precision oncology approach, several expert-curated knowledge bases have emerged[97], which annotate published information on the clinical implication of specific molecular alterations. However, a universal definition of ‘actionable alteration’ is still lacking. Furthermore, accumulating data indicate that significant genomic heterogeneity exists in a given tumor sample[98], making target selection difficult. Consequently, it is not surprising that the targeted approaches against BTC have not produced robust responses thus far. These practical hurdles will need to be overcome to exploit the knowledge of precision medicine fully.

  1. A narrative review may be a disadvantage in the field of biomarkers such as ctDNA (ctDNA is also discussed in in the text) due to potential heterogeneity of cohorts, use (or not) of adequate controls, quality material used, different techniques/platsforms used (with varying sensitivity/specificity) etc. A comment on these limitations in the text would be advised.

The following text has been added in section 4(190-196) to address this critique.

Although promising, ct-DNA based profiling of BTC has several limitations related to the technology itself and its applicability. Wide variations exist in the preanalytical variables, assay characteristics (PCR-based versus NGS- based techniques), bioinformatic analysis methods, sensitivity, and specificity[37]. These factors pose significant challenges in the interpretation of clinical trial data utilizing different ctDNA platforms. Furthermore, robust prospective clinical trial data are needed to validate the clinical utility of ctDNA in the management of advanced BTC before ctDNA-based profiling can be widely accepted in routine clinical practice.

  1. The term ‘advanced BTC’ may implicate a very different tumor load and well known variety in mutational status, when referring to iCCA, pCCA or dCCA. Although the different anatomical subtypes are mentioned in the text (Table 1), the structure of text of this review focusses on BTC in general. I would suggest to try to make a distinction (when available) between the literature on iCCA (probably larger tumors) and a separate description of pCCA/dCCA (often smaller) and different mutations.

We have not emphasized on anatomic subtypes of advanced BTC in the discussion of targeted therapy so much since the treatment of advanced BTC is increasingly dictated by the driver genomic alteration rather than the anatomic subtype. We have, however, edited the manuscript to address this critique. A specific anatomic subtype of BTC highlighted whenever data were available ( Table 2, Table 3, Lines 226,300,320,333-334,351)

  1. Within the intrahepatic CCA, we distinguish small duct and large duct intrahepatic cholangiocarcinomas with different mutations described within these subtypes. Is it possible to discuss in more detail these subtypes and/or the (ir)relevance of morphology versus mutational status in the text?

This is a constructive suggestion. We have added the following text in section 3 (line 111-115).

Accumulating data indicate that IDH1/2 and other frequent mutations have distinct clinical and prognostic implications in large duct versus small duct subtypes of iCCA. IDH1/2 mutations portend a favorable prognosis in small duct type of iCCA, but not in the large duct type, as reported in a study by Ma et al.[29]. Furthermore, BAP1 expression loss correlated with favorable prognosis only in large duct type of iCCA.

6.In line 131-132 is it stated that ctDNA-based profiling is particularly useful in BTCs, given that tumor biopsy samples are often inadequate for sequencing. The authors may consider to adress this topic (problems with biopsy) in more detail. It is a well known problem in CCA that the tumor tissue/tumor cells maybe difficult to obtain, even in the non-curative setting. What about the clinical feasibility to detect targetable mutations on biopsy/brush? And is tumor heterogeneity a relevant issue in tissue?

 The following text has been added in section 4 (138-147) to address this critique –

Tissue biopsies are often difficult to obtain, especially in extrahepatic and hilar CCA. In one meta-analysis of 16 studies, the diagnostic sensitivity of Endoscopic Retrograde Cholangiopancreatography (ERCP)-guided brushings was approximately 42%[33]. Moreover, endoscopy-based procedures are associated with various risks inherent in invasive procedures, including acute pancreatitis, which occurs in approximately 2.6% of patients undergoing ERCP. Emerging data suggest that intratumor heterogeneity and clonal evolution are inherent to malignant tumors, often driven by systemic therapies[34]. Tissue biopsy may be of limited use due to intratumor heterogeneity and repeated tissue biopsies to monitor clonal evolution are not feasible for most patients. Conversely, ctDNA analysis can be performed in peripheral blood samples obtained by outpatient phlebotomy, a minimally invasive procedure that can be repeated as needed. Consequently, ctDNA-based profiling of advanced BTC is gaining credence.

  1. Line 143-147 descibes ctDNA in 3 patients with FGFR mutations and ‘advanced BTC’. These tumors are probably iCCA. What about therapeutic monitoring with ctDNA in the context of pCCA/dCCA?

We have edited the text to indicate that these 3 patients are patients with advanced iCCA. We did not find a similar report with patients of hilar or distal CCA after an extensive search of the literature. We have also reached out to several experts, including Dr. Kabir Mody (Mayo Clinic, Florida) and Dr. P.M Kasi (University of Iowa, IA). They also confirmed that they have not come across such a report.

  1. From line 147-154 thA)e largest study is described with 70% iCCA: was it possible to separately evaluate value of ctDNA in the 70% of iCCA cases versus other localisations?

We have added the following text in section 4 (line 176-180) to address this critique-

Besides, this study demonstrated that the overall landscape of alterations was different in iCCA versus eCCA/GBC. Therapeutically relevant alterations were more frequent in iCCA compared to eCCA/GBC, with the FGFR2 alteration being the most frequent (7%) in patients with iCCA. ERBB2 alteration was exclusively found in eCCA. The ARID1A alteration was more common in eCCA, and CDK6, APC, and SMAD4 alterations were more common in GBC.

The study reported by Ettrich et al. showed that the mutational profile of the 23 available blood-tumor pairs was concordant in 74% of patients, with a higher concordance rate in iCCA( 92%) versus eCCA (55%)[37].

  1. How strong is the evidence for value of ctDNA in iCCA versus other anatomical localisations for instance?

 This critique is addressed in the previous section (point 8).

  1. Line 349-350: “All patients on targeted therapies eventually develop treatment resistance, which calls for a repeat genomic profiling at the time of progression.” Does this strong statement follow from this review? Is it validated for BTC, including value of repeat genomic profiling ( a few exemples are mentioned in the text, such as FGFR)?

We appreciate this critique. After careful consideration, we have decided to modify this statement as follows (line 403-404):

All patients on targeted therapies eventually develop treatment resistance, and a repeat genomic profiling at the time of progression may be helpful.

We look forward to receiving your response.

Sincerely,

Amit Mahipal.

Round 2

Reviewer 3 Report

Thanks to the authors for adressing my previous comments. The manuscript really improved and is acceptable in its current form.

Just as message, a quite similar review was just published in J Hepatol July 2020: Molecular targeted therapies: Ready for "prime time" in biliary tract cancer. Lamarca A, Barriuso J, McNamara MG, Valle JW.Lamarca A, et al. J Hepatol. 2020 Jul;73(1):170-185